# Synergistic Strengthening of Mechanical Properties and Electromagnetic Interference Shielding Performance of Carbon Nanotubes (CNTs) Reinforced Magnesium Matrix Composites by CNTs Induced Laminated Structure

**DOI:** 10.3390/ma15010300

**Published:** 2021-12-31

**Authors:** Zhenming Sun, Hailong Shi, Xiaoshi Hu, Mufu Yan, Xiaojun Wang

**Affiliations:** 1State Key Laboratory of Advanced Welding and Joining, Harbin Institute of Technology, Harbin 150001, China; sunzhenming611@163.com (Z.S.); huxiaoshi@hit.edu.cn (X.H.); 2Harbin Institute of Technology, School of Materials Science and Engineering, Harbin 150001, China; yanmufu@hit.edu.cn

**Keywords:** CNTs/Mg composite, laminated structure, mechanical properties, electromagnetic interference shielding effectiveness, synergistic strengthening

## Abstract

In this study, we reported a laminated CNTs/Mg composite fabricated by spray-deposition and subsequent hot-press sintering, which realized simultaneous enhancement effects on strength and electromagnetic interference (EMI) shielding effectiveness (SE) by the introduced CNTs and CNT induced laminated ‘Mg-CNT-Mg’ structure. It was found that the CNTs/Mg composite with 0.5 wt.% CNTs not only exhibited excellent strength-toughness combination but also achieved a high EMI SE of 58 dB. The CNTs increased the strength of the composites mainly by the thermal expansion mismatch strengthening and blocking dislocation movements. As for toughness enhancement, CNTs induced laminated structure redistributes the local strain effectively and alleviates the strain localization during the deformation process. Moreover, it could also hinder the crack propagation and cause crack deflection, which resulted in an increment of the required energy for the failure of CNTs/Mg composites. Surprisingly, because of the laminated structure induced by introducing CNTs, the composite also exhibited an outperforming EMI SE in the X band (8.2–12.4 GHz). The strong interactions between the laminated ‘Mg-CNT-Mg’ structure and the incident electromagnetic waves are responsible for the increased absorption of the electromagnetic radiation. The lightweight CNTs/Mg composite with outstanding mechanical properties and simultaneously increased EMI performance could be employed as shell materials for electronic packaging components or electromagnetic absorbers.

## 1. Introduction

With the rapid development of 5G telecommunication, the high speed of wireless communications, and the widespread application of portable wearable electronic devices, electromagnetic (EM) radiation has turned into a thorny global problem due to its interference to communication signals and harmful effects on human health [1,2,3,4,5]. Thus, EMI shielding has attracted much attention in recent years. The research on EMI shielding was mainly conducted among the X-band frequency range (8.2–12.4 GHz), which was crucial for the shielding performance in the military and telecommunication, such as radar and satellite phones [6,7,8]. As for the shielding materials, which are primarily applied as shells to protect electronic weapons or equipment from electromagnetic interference. As a result, the shielding materials should possess excellent mechanical properties and outstanding shielding performance at the same time [9,10,11], which remains challenging.

At present, the most widely used EMI shielding materials are conductive polymer-based composites (CPCs) and metal-based materials [12,13,14]. CPCs have been enthusiastically popular due to their advantages of low cost, lightweight and easy processability [15,16,17]. Several investigators prepared CPCs with high EMI shielding performance by adding carbon-based conductive fillers, such as carbon nanotubes, graphene, reduced graphene oxide, carbon fibers and bulk graphite [18,19,20,21,22,23,24,25,26]. However, the development of CPCs is often restricted by their relatively poor mechanical properties. Compared with the CPCs, metal-based shielding materials possess superior mechanical properties, but the drawback of heavy weight limits their application seriously. It is well known that magnesium alloy is not only the lightest metallic structure material but also performs advantages of high specific strength and high specific stiffness [27,28,29,30,31]. Moreover, extensive research has improved the EMI shielding effectiveness of magnesium alloy by alloying [32,33]. However, the EMI shielding mechanism mainly relied on reflection rather than absorption, which would produce secondary EM pollution and potential harm to humans. Besides, the EMI SE of magnesium alloy could only be improved during the low-frequency range through alloying.

Recently, it has been reported that the EMI shielding performance of metallic materials could be enhanced by the construction of internal structures [34,35,36,37]. Many studies have prepared metallic coatings (Co, Ag, Cu, Ni) into porous films with spherical internal structures, which realized a significant enhancement of EMI SE compared to the traditional dense metallic film [34,35]. Besides, it has been reported that the Cu film with the layer-by-layer assembly of Cu nanosheets exhibited better ultimate EMI performance than the bulk Cu by enhanced internal reflections [36]. It was revealed that the configurational approach mainly engenders multiple internal reflections by means of expanding the EM transmission path inside the shielding material. As a result, the incident EM is dissipated and predominantly absorbed by the shielding material.

However, although the internal structure design can generate effective EMI shielding performance, its effects on mechanical properties should also be considered. The spherical internal structure possesses an advantage in improving the EMI absorption effectiveness, but the porous feature would seriously affect the mechanical properties of magnesium alloy [37]. Zhang et al. [38] obtained outstanding electromagnetic interference (EMI) shielding effectiveness (SE) of CNTs/Mg composites prepared by accumulative roll bonding (ARB). Regrettably, the authors only discussed the electromagnetic shielding mechanism in the paper, and did not address the mechanical properties of the CNTs/Mg composites. Compared with the spherical internal structure, the laminated structure design has been reported that it could also present good effects on the enhancement of mechanical properties. Several researchers have prepared the micro-nano biomimetic laminated structures by assembling the Mg foils coated with carbon nanotubes (CNTs) or graphene nanosheets (GNSs), and the mechanical properties of the composites got significant improvement compared to the matrix Mg. It is worth noting that CNTs and GNSs are not only the ideal reinforcements for strengthening in metal matrix composites; they also possess excellent physical properties like excellent electrical conductivity and super EM-absorption properties [39,40,41,42]. Thus, there is potential to obtain simultaneous enhancement on EMI shielding effectiveness and mechanical properties of magnesium alloy through the laminated structure design by introducing CNTs (or GNSs) into the composites.

Motivated by the above considerations, in the present work, we prepared the CNTs reinforced Mg matrix (CNTs/Mg) laminated composites by assembling micron Mg foils covered with nanoscale CNTs. The effects of CNTs and the micro-nano ‘Mg-GNT-Mg’ laminated structure on mechanical properties and EMI shielding effectiveness of the CNTs/Mg composites during the X-band (8.2–12.4 GHz) frequency range were investigated systematically. The results showed that the enhanced mechanical properties and outstanding EMI shielding effectiveness of the CNTs/Mg composites were obtained simultaneously. Moreover, the strengthening mechanisms of the incorporated CNTs on the simultaneously enhanced mechanical properties and EMI shielding performance of the composites were analyzed systematically. The revealed mechanisms could be employed in other material systems to assist the fabrication of EMI shielding materials with good mechanical performance.

## 2. Experimental Methods

Figure 1 shows the preparation process of CNTs/Mg laminated composite, mainly including four procedures: surface treatment of CNTs, spray deposition (SPD), vacuum hot-press sintering and hot extrusion. The raw Mg foils (thickness of 50 μm) were purchased from Beijing Qianyue Nonferrous Metal Products Co., Ltd., Beijing, China. The Mg foils were prepared by cold-rolling with a thickness reduction of 30%. Table 1 shows the chemical composition of initial Mg foils.

### 2.1. Surface Treatment of Raw CNTs

Due to the nanometer size effect, there are relatively strong van der Waals forces between CNTs, which make CNTs easily entangled or agglomerated into clusters, and deteriorate the mechanical performance of the CNTs/Mg composites. Thus, the pickling treatment was employed to introduce oxygen-containing groups onto the CNT surfaces so as to generate strong electrostatic repulsive forces between the adjacent CNTs. In detail, the CNTs (diameter: 30–80 nm, length: <10 μm) were put into a mixture solution of H_2_SO_4_ and HNO_3_ (3:1 vol ratio) to introduce the oxygen-containing functional groups onto the CNT surface. After oscillating for 30 min, the resulting slurry was continuously stirred for 5 h at 70 °C and then rinsed, diluted and filtered repeatedly until the PH value of the slurry reached 7. Finally, the functionalized CNTs were dried in the vacuum furnace at 100 °C It is worth noting that the main effect of the pickling process is to introduce oxygen-containing functional groups to the surface of CNTs. In addition, the pickling process can also further remove the catalyst metal particles left on the surface during the preparation of CNTs as well as the amorphous carbon on the surface of CNTs, which has little effect on the dimensions of CNTs, so the length of CNTs after pickling remains basically the same as that of the original CNTs [40].

### 2.2. Spray Deposition and Fabrication of Laminated Units

Before the SPD process, the functionalized CNTs were added to alcohol to prepare a solution with CNTs concentration of 0.3 g/L. Then, the solution was treated by ultrasonic vibration for around 12 h to obtain a uniform dispersion of CNTs. For better interface bonding, the oxide layer on the initial Mg foil surfaces was removed by mechanical polishing with 350# sandpapers. Then, the high atomization spray gun (IWATA, Yokohama, Japan) was employed to spray CNT on the polished Mg foils with a deposition pressure of 0.3 MPa. It is noted that the content of CNTs introduced in the composites is controlled by adjusting the deposited volume of CNTs solution. Finally, the laminated units were successfully obtained by shredding the as-deposited Mg foils into rectangular blocks with the size of 2 mm × 2 mm.

### 2.3. Vacuum Hot-Press Sintering and Hot Extrusion

The obtained laminated units (2 mm × 2 mm) are firstly compressed in a graphite die (φ50 mm × 50 mm) at room temperature and then sintered at 630 °C for 6 h with the 50 MPa pressure in a vacuum hot-press furnace. Since the deposited CNTs did not completely cover the surface of Mg foil, there would always be exposed areas on Mg foil surface between the carbon nanotube bundles, and these adjacent areas on the Mg foil surface that were not covered by CNTs would be combined together by thermal diffusion during the vacuum hot pressing process, and finally sintered to form a complete bulk laminated composite. To densify the composites, the Hydraulic press (YA32-200, Tianjin, China) was carried out to perform the hot extrusion, which was conducted at 400 °C at an extrusion speed of 0.1 mm/s and an extrusion ratio of 29:1. The size of extruded composite was 6 mm × 11 mm × 300 mm, where the length parallel to the ED direction was 300 mm. Both the pure bulk Mg and the laminated pure Mg were fabricated using the identity procedure.

### 2.4. Materials Characterization

A field-emission scanning electron microscopy (SEM, SUPRA 55 SAPPHIRE, Oberkohen, Germany) was used to observe the dispersion of CNTs on the Mg foil surfaces and the fracture morphology of the CNTs/Mg composites. The microstructure of the CNTs/Mg composites along the extruded direction (ED) was analyzed by Electron Backscattered Diffraction (EBSD, SUPRA 55 SAPPHIRE, Oberkohen, Germany). The interfaces of CNTs/Mg were characterized by Transmission electron microscopy (TEM, Talos F200x, Portland, OR, USA). Digital image correlation (DIC) combined with the in-situ tensile test was applied to investigate the strain evolution during tensile deformation. The Instron 3382 tensile machine (Shimadzu, Kyoto, Japan) was employed to perform the in-situ tensile test with a tensile rate of 2 μm/s, and the DIC tensile samples possess the gauge length of 15 mm, the width of 3 mm and the thickness of 1 mm. Before the in-situ tensile test, the DIC samples were electropolished in a mixture of phosphoric acid and alcohol with a volume fraction of 3:5 for 30 s at a current of 0.5 A. Then, the fine emery was introduced and evenly dispersed on the observation surface of the DIC samples, which could be regarded as the gauge points for strain analysis. Finally, the images were collected by an optical microscope (Olympus Corporation, Tokyo, Japan) and the strain distribution statistics were calculated by a DIC-2D 2009 software. The electronic universal testing machine (Instron 5569, Boston, MA, USA) with standard ASTM: E8/E8m-13a was adopted for tensile testing with a crosshead speed of 0.5 mm/min. The sizes of tensile testing specimens possess a gauge length of 15 mm and width of 6 mm. The EMI SE of GNTs/Mg composites was measured via the waveguide method by Network Analyzer (VNA) (N5227A, Agilent, Palo Alto, CA, USA) in X-band (8.2–12.4 GHz). The rectangular-shaped CNTs/Mg composite specimens were machined to 22.86 mm × 10.16 mm × 5 mm for the EMI SE measurement. A digital conductivity meter (Sigma 2008B, Day Research Instruments, Xiamen, China) was used to measure the conductivity of specimens with a size of φ10 mm × 1 mm.

## 3. Results

### 3.1. Spray Deposition of GNTs

Figure 2 shows the scanning electron microscopy-secondary electron (SEM-SE) micrographs of CNT dispersion with different mass fractions. As shown in Figure 2a, the distinct rolling patterns along the rolling direction (RD) can be observed on the surface of the initial Mg foil. It can be seen that the CNTs were uniformly deposited on Mg foils after spray deposition (Figure 2b–d). There are many blank spaces that can still be observed on the as-deposited Mg foils with 0.15 wt.% CNTs (Figure 2b). The CNT coverage on the Mg foils surface varied with the increase in CNT deposition contents. Based on the image contrast, the Image-J software was applied to measure the CNT coverage on the surface of Mg foils, and the as-measured CNTs coverage on Mg foil surface were 33.68%, 50.45% and 70.28%, respectively, corresponding to CNTs mass fraction of 0.15%, 0.5% and 0.7%. Notably, stacked CNTs were observed on the Mg foil surfaces when the CNT mass fraction increased to 0.7 wt.%. This kind of CNT agglomeration should be avoided since the redundant CNTs could hinder the bonding of adjacent Mg foils, which is unfavorable to the mechanical properties of the composites.

### 3.2. Microstructure of Pure Mg and CNTs/Mg Laminated Composites

Figure 3 shows the microstructure of as-extruded laminated CNTs/Mg composite. It is seen that the micro-nano laminated structure comprising micron Mg layer and nanoscale CNT layer was successfully architected in this work. After hot extrusion, the laminated structure transformed the equiaxed grains of bulk Mg into brick-like rectangular grains. The Mg layer direction was parallel to the extrusion direction (ED), as shown in Figure 3a,b. Notably, the thickness of the initial Mg foil was reduced from 50 μm to about 20 μm after hot extrusion deformation, as shown in Figure 3c. No obvious impurities and defects were observed between the adjacent interlayers, implying that the Mg-CNT interface bonded well through the preparation method used in this work, as shown in Figure 3d. The high-resolution transmission electron microscopy (HRTEM) was carried out to further characterize the Mg-CNT interface, as shown in Figure 3e. The HRTEM image also demonstrated that there are few impurities and defects at the interface, which could strengthen the Mg-CNT interface.

### 3.3. Mechanical Properties of Mg and CNTs/Mg Composites

The tensile stress-strain curves of hot-extruded bulk Mg, laminated Mg and the CNTs/Mg composites are shown in Figure 4. The specific values of tensile properties are displayed in Table 2. Compared with the pure bulk Mg, the laminated Mg prepared by alternately stacking the pure Mg foils exhibited better strength and ductility, highlighting the advantages of the laminated structure for relieving the conflict between strength and ductility. When the CNT content was 0.15 wt.%, the yield strength of laminated CNTs/Mg composites was slightly improved compared with the laminated pure Mg. Notably, when increased the content of CNTs to 0.5 wt.%, the yield strength of the laminated CNTs/Mg composite increased from 88 MPa (bulk Mg) to 130 MPa. Meanwhile, a larger elongation to failure of 12.1% was obtained, which ulteriorly illustrated that the laminated structural design could benefit both the strength and ductility improvement. However, the laminated CNTs/Mg composite with 0.70 wt.% CNTs exhibited a sharply dropped elongation to failure of 3.2%. This may be caused by the CNT clusters which formed during the spray deposition process. The stacked CNTs on Mg foil surfaces left little space for adjacent Mg foils to bond, which weakened the interlayer bonding strength and also acted as the microcrack nucleation sources in the composites.

### 3.4. EMI Shielding Performance of Pure Mg and the CNTs/Mg Composites

The ability to attenuate the power of electromagnetic waves of the laminated CNTs/Mg composites is represented by EMI SE. The EMI SE of the total shielding effectiveness (SE_T_) is the sum of absorption (SE_A_), reflection (SE_R_) and multiple internal reflections (SE_M_). Notably, SE_M_ could be ignored when SE_T_ > 15 dB according to Schelkunoff’s theory [43]. The EMI SE could be calculated by the scattering parameters (*S*_11_ and *S*_21_) that are measured from the network vector analyzer, as shown in Equations (1)–(6) [44]:(1)$$R=\;{\left|S_{11}\right|}^{2}={\left|S_{22}\right|}^{2}$$
(2)$$T=\;{\left|S_{12}\right|}^{2}={\left|S_{21}\right|}^{2}$$
(3)A= 1−R−T
(4)$${\mathrm{SE}}_{R}=\;10\mathrm{lg}\frac{1}{1-R}$$
(5)$${\mathrm{SE}}_{A}=\;10\mathrm{lg}\frac{1-R}{T}$$
SE_T_ = SE_R_ + SE_A_(6)
where *R*, *T*, and A are the power coefficients of reflectivity (*R*), transmissivity (*T*), and absorptivity (A), respectively.

Figure 5 shows the EMI shielding effectiveness of different samples in the X-band. As displayed in Figure 5a–c, the SE_T_ value of the laminated pure Mg (~40 dB) was about twice that of pure bulk Mg (~20 dB). Notably, the SE_T_ value enhanced significantly because of the introduced laminated structure while the SE_R_ scarcely changed. This infers that the as-designed laminated structure in the pure bulk Mg is strongly related to the enhancement of absorption, which is different from the traditional metals that mainly rely on the reflection mechanism to shield electromagnetic waves [12,44]. The same phenomenon also occurred with the increased CNT content that the most SE_T_ improvement derived from the increased SE_A_, while the SE_R_ stayed almost the same (seen in Figure 5e–f). Furthermore, the SE_T_ values of CNTs/Mg composites were gradually improved with the increased content of the introduced CNTs, as shown in Figure 5d. However, the SE_A_ of the laminated CNTs/Mg composites did not maintain a linear increasing relationship with the CNT content (Figure 5e). As the CNT content increased from 0.15 wt.% to 0.5 wt.%, the corresponding SE_A_ values increased from ~12 to ~48 dB. Nevertheless, when further increasing the CNT content, the SE_A_ of the CNTs/Mg composites remained constant at a certain level. Thus, by introducing the CNTs, the SE_T_ of CNTs/Mg composites reached a maximum of about 58 dB with a CNT content of 0.5 wt.%. Because the mechanical properties of CNTs/Mg composites peaked at the same CNT content (0.5 wt.%), it can be concluded that we realized the synergistic enhancement of mechanical performance and EMI shielding properties through the incorporation of CNTs in this work.

## 4. Discussion

As mentioned above, the mechanical properties and EMI shielding performance of the laminated CNTs/Mg composites achieved simultaneous improvement in this work. The strengthening mechanisms for the two parts are systematically analyzed and discussed in the following sections, respectively.

### 4.1. Strengthening Mechanisms of the Laminated CNTs/Mg Composites

For traditional metal matrix composites (MMCs), it is well known that its strengthening mechanism mainly includes four aspects: grain refinement strengthening ($\Delta \sigma_{Hall-petch}$), load-transfer strengthening ($\Delta \sigma_{LT}$), dislocation strengthening ($\Delta \sigma_{CTE}$) by the coefficients of thermal expansion mismatch and Orowan strengthening ($\Delta \sigma_{\;Orowan}$) [28,31,45,46]. However, due to the structural architecture in this work, the strengthening mechanisms differ from the traditional MMCs and surpass the simple mixture rule [47]. Notably, since the induced CNTs were dispersed on Mg foil surfaces firstly and finally distributed at grain boundaries, the Orowan strengthening can be ignored in this work.

#### 4.1.1. Grain Refinement Mechanism

After the hot extrusion, the interlayer spacing between adjacent CNT layers was reduced to about 20 μm, as shown in Figure 3c. Due to the laminated structure induced by the CNT introduction, grain growth along the CNT nanolayer direction was free while restricted perpendicular to the interlayer direction, as shown in Figure 3b. Besides, the high-strength interlayer CNTs resulted in dynamic recrystallization during the hot-extrusion process. They generated smaller grains around the CNT layer, while the grains away from the CNT layer were tinnily refined. Finally, the bimodal microstructure was observed on the ED-TD plane of CNTs/Mg composites. The grain size of bulk Mg and 0.5 wt.% CNTs/Mg composites were statistically measured from the EBSD results, as shown in Figure 6. The results show that the average grain size of bulk Mg was 34 μm, while the CNTs/Mg composites possess the bimodal microstructure, and the average size of large and small grains was 33.4 μm and 18.9 μm, respectively. Thus, grain refinement strengthening for the laminate CNTs/Mg composites prepared in this work could be expressed by the following modified Hall–Petch relationship in Equation (7) [48]:(7)$$\Delta \sigma_{Hall-Petch}=f_{l}K\left(d_{l}^{-0.5}-d_{m}^{-0.5}\right)+f_{s}K\left(d_{s}^{-0.5}-d_{m}^{-0.5}\right)$$
where *f_l_* and *f_s_* represent the volume fraction of large and small grains in the CNTs/Mg composites, are 93.2% and 6.8%, respectively; *d_l_* and *d_s_* are the average grain sizes of the corresponding large and small grains in the composites, respectively; *d_m_* is the average grain size of bulk Mg; *K* is a constant of 280 MPa·μm^1/2^ for Mg.

#### 4.1.2. Load-Transfer Strengthening

As for load transfer strengthening, the higher CNT content and good interfacial bonding mean that more loads can be efficiently transferred from the matrix to the reinforcements. As shown in Figure 2b,c, CNTs were evenly deposited on the surface of Mg foils. However, the formed CNT films were not continuous and compact in this work, which left empty areas that ensured the excellent connection between adjacent Mg foils through thermal diffusion and formed a bridge-like Mg pile during the deformation process, as shown in Figure 7a. It also implies that the CNTs introduced between the adjacent Mg foils bonded well with the Mg matrix and could effectively bear the load from the Mg matrix (seen in Figure 3d). As shown in Figure 7b,c, the pulled-out CNTs and bridge-like CNTs were observed at the fracture surfaces. Both of them demonstrated that the CNTs could effectively bear the load from the interface and strengthen the composites. In addition, compared with other reinforcements, CNTs exhibit extremely high specific areas and could efficiently fulfill the load transfer strengthening effect. The load transfer mechanism has been widely explained by the shear lag model, and the following equation (Equation (8)) of the modified shear lag model could be applied to calculate the contribution of load transfer to the enhancement of the yield strength (YS) of the composites [49]:(8)$$\Delta \sigma_{LT}=\sigma_{m}\frac{V_{f}S}{4}$$
where $\sigma_{LT}$ and $\sigma_{m}$ are yield strength of the CNTs/Mg composites and Mg matrix, respectively; *V_f_* is the volume fraction of CNTs in the composites and *S* is the aspect ratio of CNTs.

#### 4.1.3. Dislocation Strengthening

During the sintering, cooling and deformation processes of composites, the geometrically necessary dislocation (GND) generated at the CNT-Mg interface due to the thermal expansion mismatch and incompatible plastic deformation ability between CNT layers and Mg layers. What is more, the CNT layers also accumulated dislocations by inhibiting the dislocation movements during tension, as shown in Figure 8.

The dislocation strengthening generated due to the incompatible thermal expansion behavior could be calculated by Equation (9) [41]:(9)$$\Delta \sigma_{CTE}=AGb{\left[\frac{12\Delta \alpha \Delta TV_{f}}{bd_{c}}\right]}^{1/2}$$
where *A* is a constant (1.25) [48]; *G* is the shear modulus of the Mg matrix (1.67 × 10^4^ MPa); b is the Burgers vector of the Mg matrix (3.20 × 10^−10^ m) [46]; Δ*T* is the difference between the extrusion and test temperature; Δα is the difference between the CTE of Mg matrix and CNTs (2.4 × 10^−5^ K^−1^) [46]; *V_f_* is the volume fraction of CNTs; $d_{c}$ is the effective diameter of CNTs.

The calculated contributions of the mechanisms mentioned above to enhance the strength of 0.5 wt.% CNTs/Mg composite are displayed in Figure 9. It could be seen that the theoretical enhanced strength (30.8 MPa) is almost equal to the experimental value (31 MPa). By comparison, thermal expansion mismatch strengthening (22.1 MPa) is the predominant strengthening mechanism, while the load transfer strengthening (7.2 MPa) provided a small part of the enhancement in the total strength. Because the volume fraction of the refined grains is small and only appeared near the interlayer, the enhancement by grain refinement (1.5 MPa) could be negligible.

### 4.2. Toughening Mechanisms of the Laminated CNTs/Mg Composites

For traditional MMCs, reinforcements usually decrease the ductility of composites [38,40,45]. However, the CNTs induced laminated structure played a significant role in balancing the strength and ductility of the composites in this work. The digital image correlation (DIC) was employed to observe the local strain evolution during the tensile deformation process. Figure 10 shows the visualized local strain maps ($\varepsilon_{xx}$) parallel to the loading direction on the ED-TD plane. The strain localization could be clearly observed in bulk Mg. With the increase in the tensile strain, the strain concentration got increasingly severe. Compared with the bulk Mg, the laminated 0.5 wt.% CNTs/Mg composites here presented homogeneous deformation. At low deformation strain of 0.5%, there was an obvious subregion of tensile strain and compressive strain in the composites, which means the emergence of strain localization. With the gradual increase in deformation to 2%, the local compressive strain faded away little by little, and the tensile strain became dominant at more regions, which made the strain distribution more uniform in the composite, as shown in Figure 11a. From the observation of local strain evolution at the high strain range in Figure 11b, only the tensile strain was observed in the laminated CNTs/Mg composite. When further increasing the deformation strain, it can be seen that the plastic deformation was not constrained in an individual layer but was extended to a larger region. In other words, more areas participated in the deformation, and the strain concentration of the laminated composite was relieved. Consequently, the laminated structure induced by the introduction of CNTs could effectively inhibit the trend of strain localization and work for redistributing the local strain during the deformation process.

Figure 12a–c shows the SEM images of fracture surfaces of 0.5 wt.% CNTs/Mg composites at different tensile strains. When the tensile strain reached 6%, a large number of microcracks were observed to generate evenly around the interface, which also evidenced the relatively stable plastic deformation in the laminated CNTs/Mg composites (seen in Figure 12b). When the tensile strain reached 12%, it can be seen that the previously observed microcracks grew up at the former place but the interface hindered its propagation. Moreover, the crack deflection was also observed on the surface of the fracture, as shown in Figure 12c. The SEM image of the side fracture of 0.5 wt.% CNTs/Mg composites also proved the trajectory of crack deflection, as shown in Figure 12d. The schematic diagram of the fracture model in Figure 12f depicted the crack deflection at the fracture. By means of crack deflection, the path of crack propagation was extended, resulting in increased required energy for crack propagation and enhanced the toughness of the composites. Besides, with the development of macroscopic cracks, the fracture of the composite is gradually controlled by the incremental failure behavior of the individual CNTs, such as the pull-out and bridged CNTs in Figure 7. The failures of carbon nanotubes occur mainly along the length of the CNTs embedded in the Mg matrix, which also plays a role in the enhancement of toughness for the composites [42,50,51]. Differently, the bulk Mg tended to generate localized strain during tensile deformation and failed with a relatively smooth fracture, as shown in Figure 12e. The corresponding schematic diagram of the fracture model is displayed in Figure 12f.

Figure 13a–f shows the EBSD results of the bulk Mg and 0.5 wt.% CNTs/Mg composites. From the scattered pole figures in Figure 13a,b, it can be seen that there was a strong texture in bulk Mg, while the texture intensity of the CNTs/Mg composites was weak. Compared with the bulk Mg, the CNTs/Mg composites exhibited more scattered grain orientations due to the microstructure evolution caused by the introduced layered structure. It is well known that the basal slip of dislocations is the predominant deformation mechanism of Mg at room temperature. Figure 13c,e demonstrated the Schmid factors for the basal slip system along the ED direction of the bulk Mg. The lower value of the Schmid factor means that it is not favored during the deformation process. Different from the bulk Mg, the CNTs/Mg composites exhibited a higher value of Schmid factors, as shown in Figure 13d,f, which implied a favorable plastic deformation capacity of grains. The main reason for the different Schmid factor results between bulk Mg and CNTs/Mg composites is the laminated structure induced by CNTs. During the hot extrusion process of bulk Mg, the unconstrained grains could easily rotate along the ED direction. In contrast, the grains of CNTs/Mg composites were strongly constrained due to the introduced laminated structure and were hard to rotate freely. Thus, the texture intensity of CNTs/Mg composites was seriously reduced, which effectively enhanced the dislocation slipping ability. Moreover, in the 0.5 wt.% CNTs/Mg composites, obvious slip bands were observed on the ED-TD plane of the fracture surfaces at the tensile strain of 12%, which also demonstrated the favorable plastic deformation ability of the composites (seen in Figure 13g). As stated above, the laminated structure induced by introducing CNTs could strengthen the ductility of CNTs/Mg composites.

### 4.3. EMI Mechanisms of the Laminated CNTs/Mg Composites

The laminated CNTs/Mg composites prepared in this work exhibited complicated EMI mechanisms compared to traditional homogeneous conducting metals due to the laminated structure induced by introducing CNTs and interlayer CNTs. Previous studies on EMI mechanisms have indicated that the internal structure and electrical conductivity of shielding materials play an important role in improving the absorption and reflection of incident electromagnetic waves, respectively [13]. In general, the highly conductive shielding materials exhibit high electromagnetic shielding effectiveness because of the forceful reflection of incident electromagnetic waves through their surfaces. The structural design inside the shielding materials is expected to dissipate the energy of the electromagnetic waves by increasing the multiple internal reflection paths of the electromagnetic waves [36]. As shown in Figure 14, the micro-nano laminated structure was architected by layer-by-layer stacking the Mg microlayers and CNT nanolayers, which strongly contributed to the multiple internal reflections of incident electromagnetic waves and significantly enhanced the absorption of electromagnetic waves for the laminated CNTs/Mg composites. Therefore, although the electrical conductivity of laminated pure Mg and CNTs/Mg composites all decreased compared with the pure bulk Mg (seen in Figure 15), the SE_A_ values were still obviously elevated (shown in Figure 5). When the CNT content increased from 0.15 wt.% to 0.5 wt.%, the corresponding SE_A_ values gradually increased to about 48 dB (seen in Figure 5). It implied that the laminated architecture and the introduction of CNTs both contributed to enhancing the absorption of electromagnetic waves. It has been proven that the interface between two kinds of conductive materials whic possess relatively different impedance could effectively reflect the electromagnetic waves [36]. Thus, as the content of CNTs gradually increased, the electric conductivity of CNT nanolayers was elevated and enormously reinforced the reflection intensity of electromagnetic waves by the internal micro-nano interfaces. Moreover, more CNTs would generate more micro-nano interfaces and could expand the multiple internal reflection paths for the electromagnetic waves. Hence, the energy of the incident electromagnetic wave was further dissipated with the increased loading of CNTs.

It is well known that vertically accelerated charged electrons are accompanied by the radiation of electromagnetic waves, which would generate traveling oscillation waves of electric and magnetic fields, and these two waves propagate at right angles as well as in phase sinusoidally to each other [52,53]. Some of the previous studies [54] have indicated that the electromagnetic waves are more likely to transmit directly through the CNTs when the CNTs are aligned in a direction parallel to the oscillating direction of the electromagnetic waves. In addition, the electromagnetic waves tend to react with the CNTs that are aligned in the other directions either by reflection or absorption. Consequently, the CNTs in different directions could be regarded as the filters of the incident electromagnetic waves.

In this work, the orientation of the CNTs between the adjacent Mg foils was random after the spray deposition process (seen in Figure 2). As shown in Figure 14, once the incident electromagnetic waves reach the CNT nanolayers, only the electromagnetic waves in one oscillating direction are allowed to transmit through the corresponding first CNT filter, and the transmitted electromagnetic waves would then be strongly blocked by the second CNT filter. Thus, with more incorporated CNTs, the thicker CNT nanolayers could block more incident electromagnetic waves, which intensified the collision between electromagnetic waves and CNTs. In other words, more reactions between the mobile charge carriers of polarized CNTs and electromagnetic waves occurred, and the energy of the electromagnetic waves was mightily dissipated.

In addition, it is noteworthy that the CNTs used in this work are multi-walled carbon nanotubes (MWNTs), which are similar to the rolled-up multi-walled graphene and exhibit a cylinder of carbon atoms arranged on a hexagonal lattice [55]. The geometrical characteristics and extremely high aspect ratio of MWNTs make it easy for the incident electromagnetic waves to be transmitted into the innermost wall of MWNTs [56]. Besides, the forceful mobile charge carriers, magnetic and electron dipoles of MWNTs can strongly react with electromagnetic waves and effectively decrease the EMI impact by means of absorption [57] so that the radiation of the electromagnetic waves is absorbed to the greatest extent between all the walls of the MWNTs.

To sum up, as the content of CNTs gradually increases, the multiple internal reflections, absorption, and dissipation of electromagnetic waves are effectively enhanced, contributing a high absorption value to the EMI SE of CNTs/Mg composites. However, when further increased the CNT content from 0.5 wt.% to 1.0 wt.%, the CNT content made no difference to the value of SE_A_ and resulted in the SE_T_ value of CNTs/Mg composites remaining stable. On the one hand, the CNT conductive network was formed in the CNT layers once the CNT content reached a saturation value, and the electrical conductivity was no longer affected by increasing the content of CNTs, as shown in Figure 15. It implied that there was no more mobile electric charge in CNT nanolayers to react with the incoming electromagnetic waves. On the other hand, the superabundant CNTs are inclined to cause CNT agglomeration, which could not offer more exposed surfaces to react with the incoming electromagnetic waves and distinctly reduced internal multiple reflections.

## 5. Conclusions

In this work, we fabricated the laminated CNTs/Mg composites by spray-deposition and subsequent hot-press sintering. The mechanical properties and EMI shielding performance of the CNTs/Mg composites were simultaneously improved because of the incorporated CNTs and the CNT-induced laminated structure. The strengthening mechanisms on the two parts were analyzed thoroughly.

The strength is predominantly enhanced from thermal expansion mismatch strengthening and load transfer strengthening. As for enhancement in ductility, the laminated structure induced by CNTs could not only redistribute the local strain and alleviate the strain localization during the deformation process but also could hinder the crack propagation and lead to crack deflection, which results in an increment of the required energy for the failure of the CNTs/Mg composites. Moreover, the texture intensity was weakened by the introduced CNT layers, which gave rise to a higher value of Schmid factor on the basal slip system {0 0 0 1} 〈11–20〉, so that the ductility of laminated CNTs/Mg composites is enhanced. Moreover, because of the laminated structure induced by the incorporated CNTs, the composites exhibit an outperforming EMI shielding efficiency in the X band (8.2–12.4 GHz). The strong interactions between CNTs and the incoming electromagnetic waves are responsible for the effective absorption of electromagnetic radiation. The lightweight CNTs/Mg composites with outstanding strength-ductility can be applied as shell materials for electronic packaging components. The revealed mechanisms could be employed in other material systems to assist the fabrication of EMI shielding materials with good mechanical performance.

## Figures and Tables

**Figure 1 materials-15-00300-f001:**
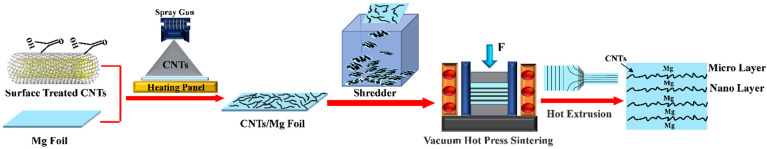
Schematic illustration for the fabrication process of CNTs/Mg laminated composite.

**Figure 2 materials-15-00300-f002:**
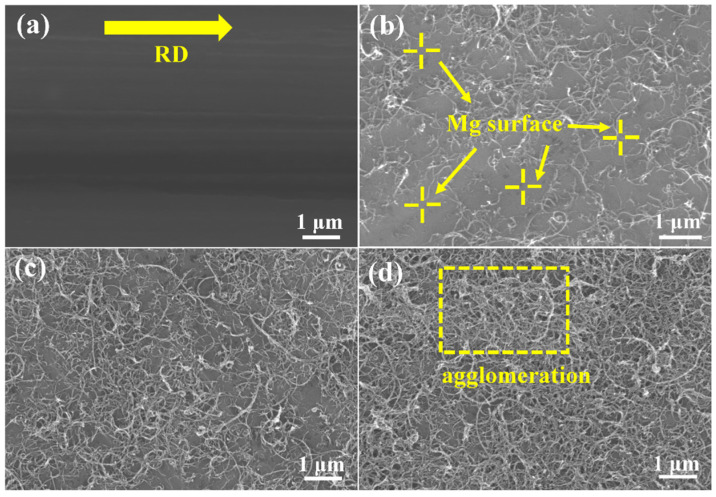
SEM-SE images of Mg foils deposited with distinct mass fraction of CNTs. (**a**) Pure Mg foil; (**b**) 0.15 wt.%; (**c**) 0.50 wt.%; (**d**) 0.70 wt.%.

**Figure 3 materials-15-00300-f003:**
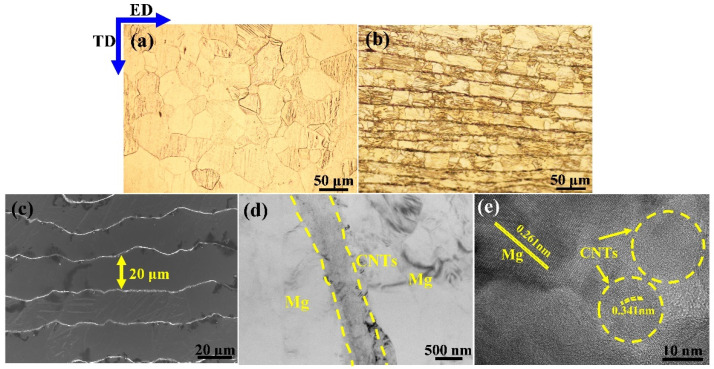
Microstructure of as-extruded laminated CNTs/Mg composite. (**a**,**b**) OM image of bulk Mg and CNTs/Mg composite on ED-TD plane, respectively; (**c**) SEM photograph of laminated CNTs/Mg composite on ED-TD plane with adjacent CNTs layers spacing ~20 μm; (**d**) TEM image from ED-TD plane of hot-extruded CNTs/Mg composite; (**e**) HRTEM image demonstrates the interface between Mg matrix and CNTs.

**Figure 4 materials-15-00300-f004:**
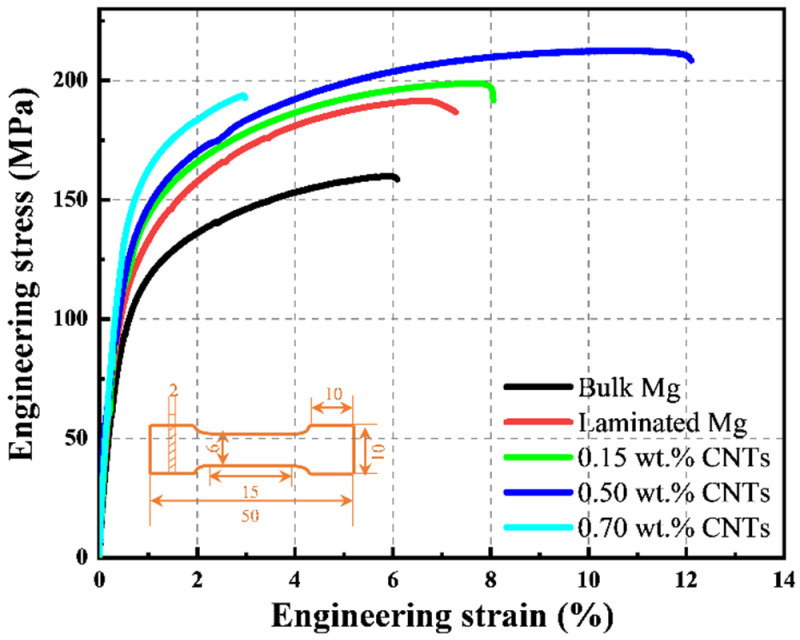
Engineering stress-strain curves of bulk Mg, laminated Mg and CNTs/Mg composites.

**Figure 5 materials-15-00300-f005:**
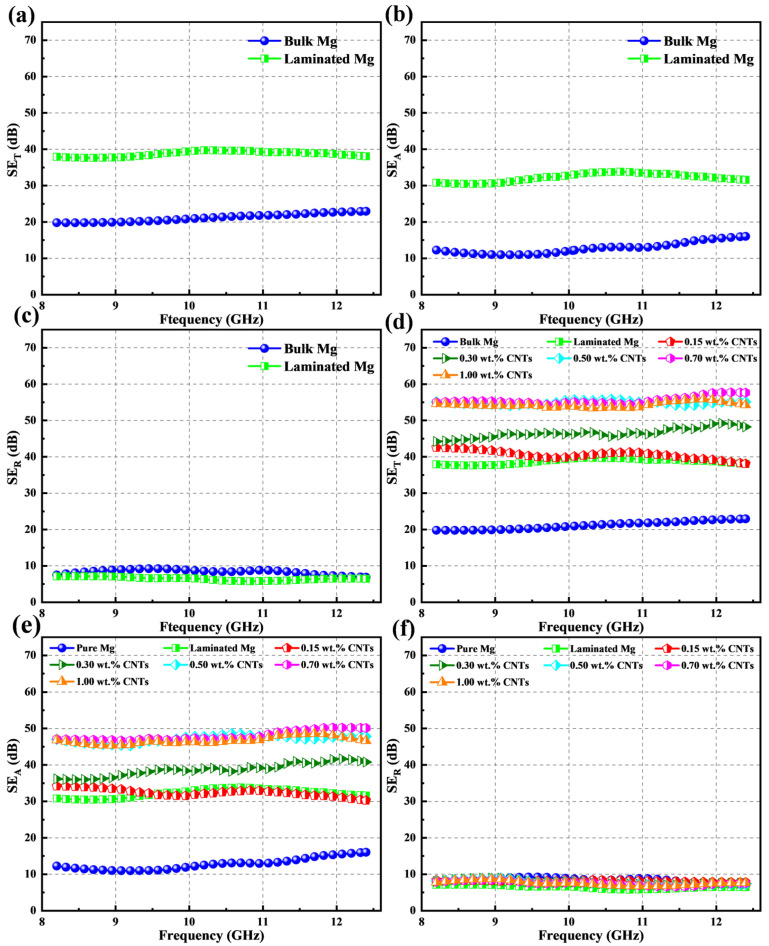
The EMI SE results of bulk Mg, laminated Mg and CNTs/Mg composites in the X-band. (**a**–**c**) total (SE_T_), absorption (SE_A_) and reflection (SE_R_) shielding effectiveness of bulk Mg and laminated Mg, respectively; (**d**–**f**) total (SE_T_), absorption (SE_A_) and reflection (SE_R_) shielding effectiveness of laminated CNTs/Mg composites with different CNT contents, respectively.

**Figure 6 materials-15-00300-f006:**
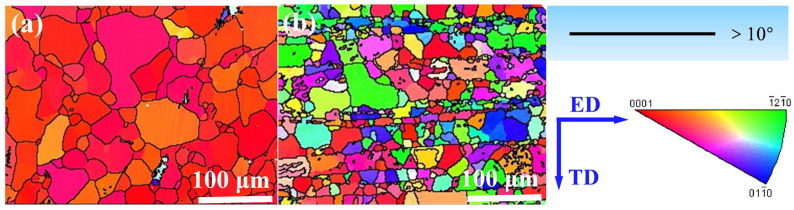
The IPFs results of bulk Mg and 0.5 wt.% CNTs/Mg composites on the ED-TD plane. (**a**) bulk Mg; (**b**) CNTs/Mg composites.

**Figure 7 materials-15-00300-f007:**
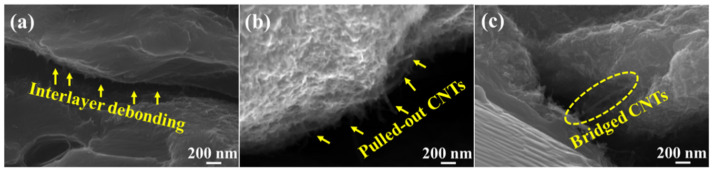
SEM-SE fracture micrographs of 0.5 wt.% CNTs/Mg composites. (**a**) interlayer debonding between adjacent Mg foils after tensile deformation; (**b**,**c**) demonstrate the pulled-out and bridged CNTs at the fractured surfaces, respectively.

**Figure 8 materials-15-00300-f008:**
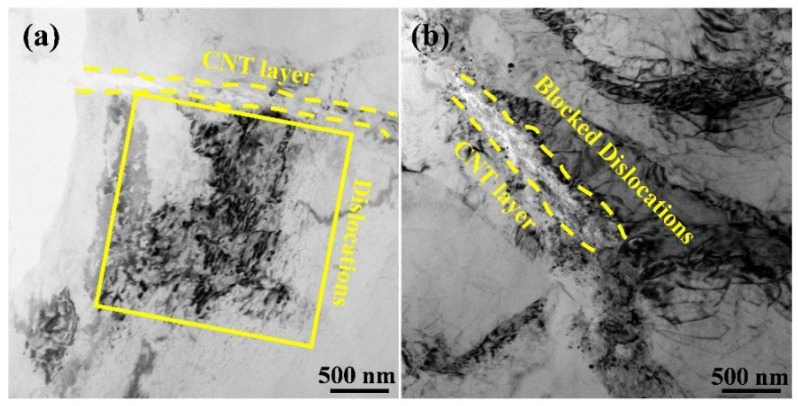
TEM images of 0.5 wt.% CNTs/Mg composites. (**a**) shows the dislocations blocked by the CNT layer after hot extrusion, highlighted by the yellow box; (**b**) accumulated dislocations around the CNT layer after tensile test. The dashed line shows the position of the CNT layer.

**Figure 9 materials-15-00300-f009:**
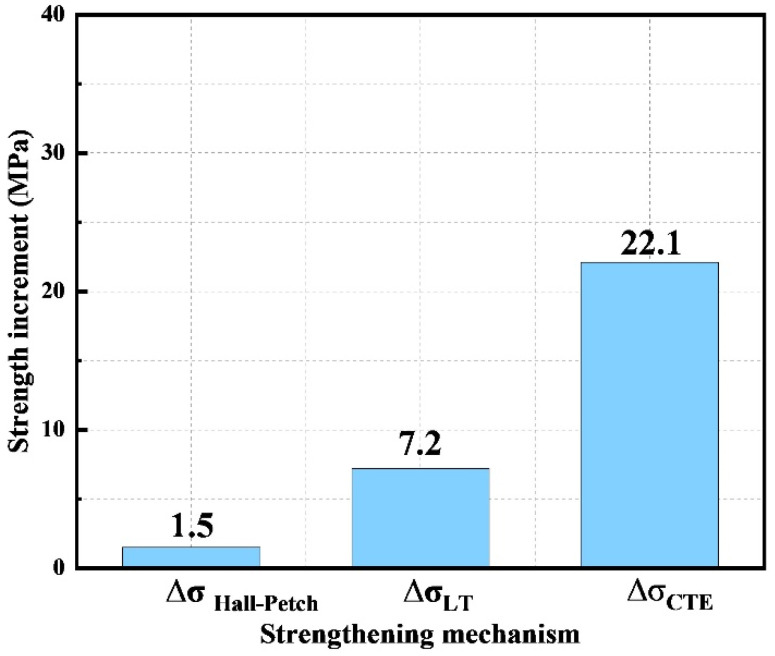
Calculated strength increment contributed by different strengthening mechanisms.

**Figure 10 materials-15-00300-f010:**
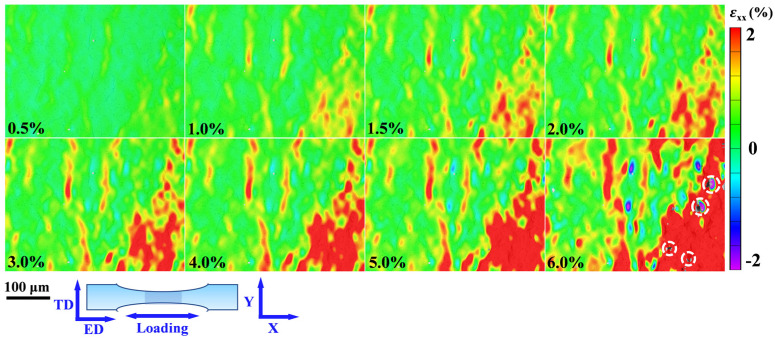
Strain evolution $\varepsilon_{xx}$ (parallel to the tensile direction) of hot-extruded bulk Mg.

**Figure 11 materials-15-00300-f011:**
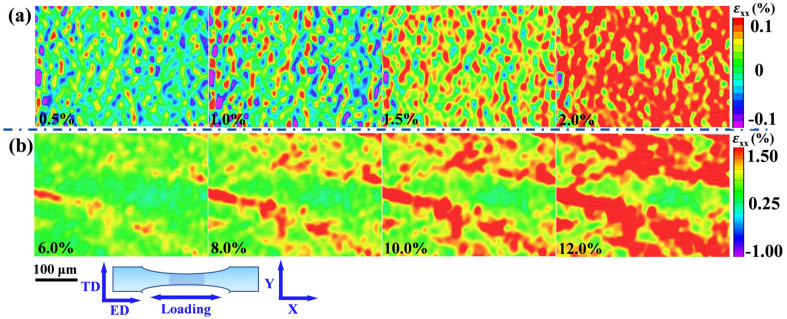
Strain evolution $\varepsilon_{xx}$ (parallel to the tensile direction) of hot-extruded laminated 0.5 wt.% CNTs/Mg composite. (**a**) Low strain range; (**b**) High strain range.

**Figure 12 materials-15-00300-f012:**
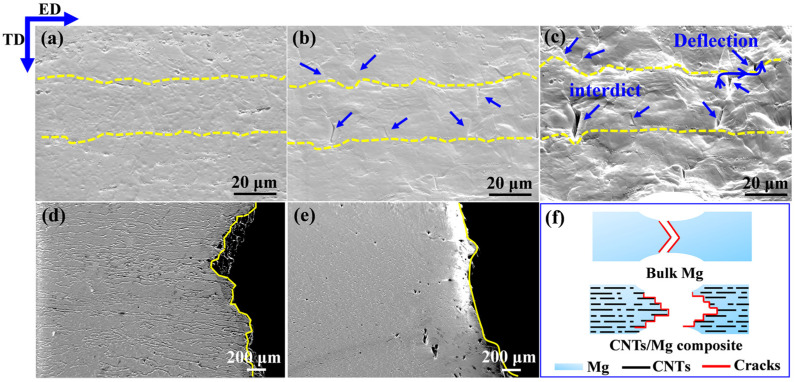
SEM images of fracture surfaces of 0.5 wt.% CNTs/Mg composites at different tensile strains. (**a**) 0%, (**b**) 6% and (**c**) 12% on the ED-TD plane, the dashed line shows the position of the CNT layer and the arrow indicates the cracks; (**d**,**e**) present the trace of crack deflection in 0.5 wt.% CNTs/Mg composites CNTs/Mg composites and bulk Mg, respectively; (**f**) displays the schematic diagram of fracture model of bulk Mg and CNTs/Mg composites, respectively.

**Figure 13 materials-15-00300-f013:**
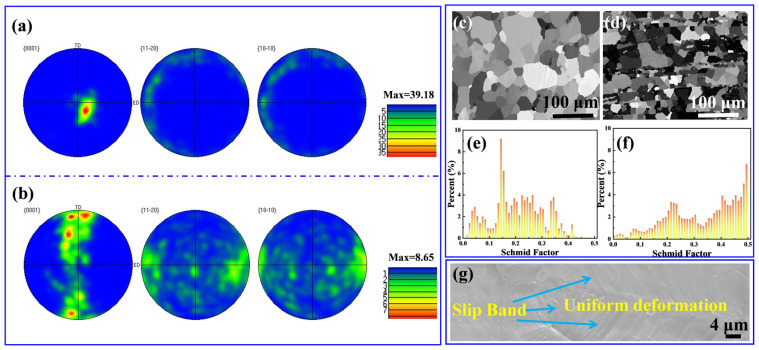
EBSD results of the bulk Mg and 0.5 wt.% CNTs/Mg composites corresponding to (**a**–**f**), (**a**,**b**) show the pole figures of bulk Mg and CNTs/Mg composites, respectively; (**c**,**d**) are the Schmid factor maps of bulk Mg and CNTs/Mg composites, respectively. The darker area means larger Schmid factor for the basal slip system {0 0 0 1} 〈11–20〉 along the ED direction; (**e**,**f**) are the value of Schmid factor distribution corresponding to (**c**,**d**); (**g**) is the SEM graph of the fracture surface of 0.5 wt.% CNTs/Mg composites with a tensile strain of 12% on ED-TD plane.

**Figure 14 materials-15-00300-f014:**
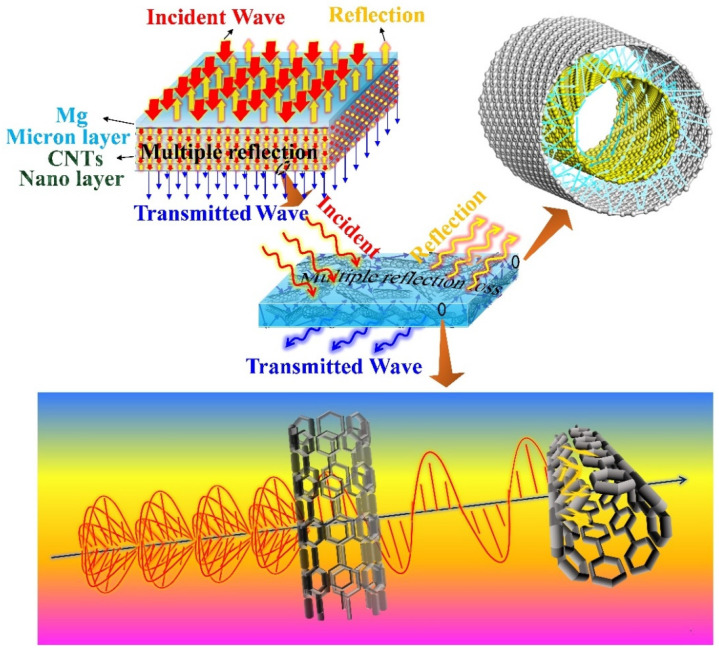
Schematic diagram of the EMI shielding mechanism in the laminated CNTs/Mg composites.

**Figure 15 materials-15-00300-f015:**
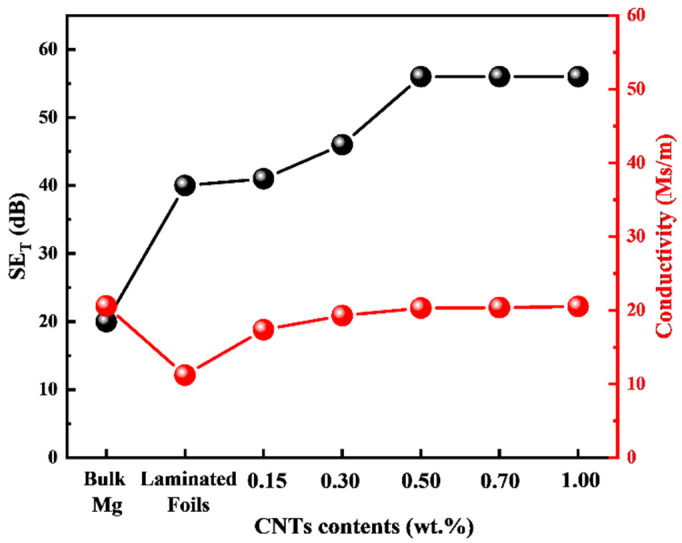
Electrical conductivity and corresponding EMI SE of the CNTs/Mg composites.

**Table 1 materials-15-00300-t001:** Chemical composition of initial Mg foils.

Alloying Element	Fe	Si	Ni	Cu	Al	Cl	Mn	Ti
Content (%)	0.004	0.005	0.0007	0.003	0.006	0.003	0.01	0.014

**Table 2 materials-15-00300-t002:** Mechanical properties of Mg and laminated CNTs/Mg composite.

Materials	YS (MPa)	UTS (MPa)	Elongation (%)
Bulk Mg	88	159	6.1
Laminated Mg	102	185	7.3
0.15 wt.% CNTs/Mg	119	198	8.0
0.50 wt.% CNTs/Mg	130	213	12.1
0.70 wt.% CNTs/Mg	137	192	3.2

## Data Availability

The data presented in this study are available upon request from the corresponding author.

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
