# Peer review of "Synergistic Strengthening of Mechanical Properties and Electromagnetic Interference Shielding Performance of Carbon Nanotubes (CNTs) Reinforced Magnesium Matrix Composites by CNTs Induced Laminated Structure"

_materials, 2021, doi:10.3390/ma15010300_

Round 1

Reviewer 1 Report

The manuscript entitled "Synergistic strengthening of mechanical properties and electromagnetic interference shielding performance of Carbon Nanotubes (CNTs) reinforced magnesium matrix composites by CNTs induced laminated structure" by Sun, Shi, Hu, Yan and Wang reports the development and mechanical/electromagnetic shielding characterisation of a CNT-magnesium metal matrix laminate. The authors found and report simultaneous optimization of mechanical and EM shielding properties of the composite at a 0.5% CNT loading. This result is important as extremely few studies suggest a common concentration of maximization of properties of independent nature. The mechanical improvement was rationalized in terms of dislocation strengthening and toughening offered by classical energy dissipation micromechanisms. The EM shielding performance in the X band, in terms of the laminated structure and the strong interactions between the tubes and EM waves. The work reflects well planned and executed research with clear target. A good number of experimental techniques were used including spray deposition, hot pressing, extrusion, DIC, SEM and TEM. The results are very interesting for the materials community, especially in view of the expansion of 5G communication. The quantification of the contribution of different mechanisms to strength (Fig. 9) is also an important finding. On the other hand, the work needs better explanations and clarifications at some points. It must aslo be corrected in grammar and syntax by a native english speaker. The main amendments required are:

1) Why where the tubes functionalized for spray deposition? The alcohol environment used for the spray solution is a well known friendly environment for CNTs.

2) Please include the dimensions of as-extruded plates.

3) Authors state that "coverage on Mg foil surface 160 were 33.68%, 50.45% and 70.28%, respectively, corresponding to CNTs mass fraction of 0.15%, 0.5% and 0.7%." How did the authors manage to adjust deposition time in a way that the resulting coverage corresponded to such round mass concentration values?

4) The manuscript mentions that "Compared with the pure bulk Mg, the laminated Mg prepared by alternately stacking the pure Mg foils exhibited better strength and toughness, highlighting the advantages of the laminated structure for relieving the conflict between strength and toughness." It is unclear why two pure specimens of the same material can respond differently in mechanical terms, solely due to lamination. Please explain in view of equal strain conditions.

5) The manuscript contains a lot of discussion on load transfer and toughening. As many of the information scattered in the text refers to the same theory, a better explanation of the overall behavior is vital. First, a good interfacial bonding is quoted, derived by TEM information. In fact, all the energy dissipation micromechanisms quoted, such as bridging and pull-out, require a moderate interface that can debond. A strong interface does not debond and promotes brittle fracture (small energy dissipation). Secondly, the authors find evidence of crack deflection. Together with debonding, bridging and pull-out, crack deflection is the classical energy dissipation toughening micromechanism for composites. The authors are advised to consult Acta Materialia 55, 2007 , pp. 83-92 and Acta Materialia 51, 2003, pp. 5359-5373 and offer a better unified discussion of the development of crack deflection and debonding due to the moderate interfacial shear strength, which in turn gives rise to the toughening micromechanisms of bridging and pull-out. Is the evidence of pull-out a sufficient proof of previous bridging?

6) Shouldn't the DIC color maps, Fig 10 and 11, be rotated vertically so that their long axis is parallel with the gauge length as pictured?

7) Authors attribute the compressive-tensile areas in Fig.11a at 0.5%, to strain localization. What is the origin of this localization? It appears to the reader that compressive fields must be the effect of thermal residual stresses, due to CTE mismatch.

8) Authors quote that their CNTs were "prepared by rolling up multi-walled graphene". It is highly unlikely that this is the case. Geometrically, CNTs can be conceived as axially-joined rolled graphene sheets, but for sure they're NOT prepared by rolling graphene. 

9) Authors quote "the mechanical properties of the composites got significant improvement compared to the matrix Mg." In the next sentence they quote "CNTs and GNSs are not only the ideal reinforcements for strength strengthening in metal matrix composites...". On what evidence CNTs and GNSs are ideal reinforcements for MMCs?

10) Please correct grammar/syntax mistakes, for example:

line 25 - The lightweight CNTs/Mg composit
line 32 - the high speed of (font size)
line 36-37 The research on EMI shielding was mainly conducted during the X-band frequency range (during is not the correct word)
line 39 And the shielding materials are primarily applied ("and" is a conjunction joining two independent phrases within the same sentence, hence sentences cannot start with "and")
line 73 Differently, the laminated (by differently maybe you mean "on the other hand"?)
line 75 coated with nanoscale carbon (nanoscale is redundant for CNTs)
line 78-79 for strength strengthening in metal matrix composites (double use of word)
line 93 were analyzed deeply (deeply does not sound correct here)
line 157-158 With the increase of CNT deposition contents, the Mg foils were covered by as-deposited continuous CNTs films by degrees. (degrees is not the correct word)
line 332 - Differently, (same as above)

Reviewer 2 Report

The presented article is devoted to the study of the mechanical properties and the electromagnetic interference shielding effectiveness of laminated Mg-based composites reinforced with 0.15-1 wt.% carbon nanotubes and produced by spray deposition followed by hot press sintering. This article is well presented and discussed and can be published after some revision based on the following comments.

1) Lines 82-83: There is a missing word in a phrase "through the laminated structure design by introducing CNTs (or ???s) into the composites".
2) Equipment for spray deposition, hot extrusion, DIC investigation and in-situ tensile test, as well as electrical conductivity measurements should be specified in the Experimental section.
3) It is not clear how the graphite die for sintering with the size of 50×50 mm was uniformly filled with shredded rectangular blocks with the size of 2×2 mm so that as a result continuous layers were formed. In addition, the shape and dimensions of the final sample after hot extrusion should be indicated.
4) "Ductility" is more preferable in comparison with "toughness" in the text.
5) The font of the legend in Figures 5 d-f and Fig. 10-11, as well as the font in Figures 13-14 should be made more visible.
6) In the text and the figure captions of the Discussion section, it is necessary to indicate the composition of the material for which the studies were carried out.
7) The article of Zhang, W.; Zhao, H.; Hu, X.; Ju, D. A Novel Processing for CNT-Reinforced Mg-Matrix Laminated Composites to Enhance the Electromagnetic Shielding Property. Coatings 2021, 11, 1030. (https://doi.org/10.3390/coatings11091030) should be referenced, and the results from this article should be compared with ones obtained in the present paper.

Reviewer 3 Report

The authors have fabricated a laminated CNTs/Mg composite by spray-deposition and subsequent hot-press sintering. The mechanical property and electromagnetic interference shielding performance were investigated. The results were interesting, which may provide insights for the fabrication of functional metal matrix composites. Here are some issues that should be addressed before the acceptance.

  1. The dimension of used CNTs is important for the strengthening. What is the length of CNTs after the surface treatment?
  2. Why the elongation of the laminated 0.5 wt.% CNTs/Mg composite is enhanced as compared with pure Mg?
  3. It seems that the CNT layers are not straightly aligned in Fig. 3c, showing a waved interface with the Mg matrix. It is possibly good for the load transfer. So why are such waved interfaces formed?
  4. Do the authors consider the effect of the oxygen-containing functional groups of CNTs on the interfacial bonding between the CNT and Mg matrix?
  5. The authors believe that the oxide layer of Mg foil is not good for the interfacial bonding between CNTs and Mg. However, it is reported that the existence of oxides may benefit to the enhanced interface bonding of the composites, doi.org/10.1016/j.msea.2021.140784; doi.org/10.1016/j.jallcom.2019.03.063; doi.org/10.1080/21663831.2020.1861120. Please clarify it.
  6. Can the carbon/carbon be directly bonded together during the hot-press sintering

Round 2

Reviewer 1 Report

Manuscript has been amended accordingly, it is clear for publication.

Reviewer 3 Report

The manuscript now can be published.